# Re-engagement and retention in HIV care after preventive default tracking in a cohort of HIV-infected patients in rural Malawi: A mixed-methods study

Jean Christophe Dimitri Suffrin[1]*, Anat Rosenthal[2], Linda Kamtsendero[1], Chiyembekezo Kachimanga[1], Fabien Munyaneza[1], Jonathan Kalua[3], Enoch Ndarama[3], Clement Trapence[1], Moses Banda Aron[1,4], Emilia Connolly[1,5,6‡], Luckson W. Dullie[1,7‡]

1 Partners In Health, Neno, Malawi, 2 Department of Health Policy and Management, Faculty of Health Sciences, Ben-Gurion University of the Negev, Be'er Sheva, Israel, 3 Ministry of Health, Neno District Hospital, Donda, Malawi, 4 Research Group Snake Bite Envenoming, Bernhard Nocht Institute for Tropical Medicine, Hamburg, Germany, 5 Division of Pediatrics, College of Medicine University of Cincinnati, Cincinnati, Ohio, United States of America, 6 Division of Hospital Medicine, Cincinnati Children Hospital Medical Center, Cincinnati, Ohio, United States of America, 7 Department of Family Medicine, School of Medicine and Oral Health, Kamuzu University of Health Sciences, Blantyre, Malawi

‡ EC and LWD are joint senior authors on this work.
* sdimitri33@gmail.com

**Data Availability Statement:** The quantitative data supporting the results of this study is uploaded as supplementary information. Qualitative data is

## Abstract

Loss-to-follow-up (LTFU) in the era of test-and-treat remains a universal challenge, especially in rural areas. To mitigate LTFU, the HIV program in Neno District, Malawi, utilizes a preventive default tracking strategy named Tracking for Retention and Client Enrollment (TRACE). We utilized a mixed-methods descriptive study of the TRACE program on patient's re-engagement and retention in care (RiC). In the quantitative arm, we utilized secondary data of HIV-infected patients in the TRACE program from January 2018 to June 2019 and analyzed patients' outcomes at 6-, 12-, and 24-months post-tracking. In the qualitative arm, we analyzed primary data from 25 semi-structured interviews. For the study period, 1028 patients were eligible with median age was 30 years, and 52% were women. We found that after tracking, 982 (96%) of patients with a 6-week missed appointment returned to care. After returning to care, 906 (88%), 864 (84%), and 839 (82%) were retained in care respectively at 6-,12-, and 24-months. In the multivariate analysis, which included all the covariates from the univariate analysis (including gender, BMI, age, and the timing of ART initiation), the results showed that RiC at 6 months was linked to WHO stage IV at the start of treatment (with an adjusted odds ratio (aOR) of 0.18; 95% confidence interval (CI) of 0.06–0.54) and commencing ART after the test-and-treat recommendation (aOR of 0.08; 95% CI: 0.06–0.18). RiC after 12 months was associated with age between 15 and 29 years (aOR = 0.18; 95%CI: 0.03–0.88), WHO stage IV (aOR = 0.12; 95%CI: 0.04–0.16) and initiating ART after test-and-treat recommendations (aOR = 0.08; 95%CI: 0.04–0.16). RiC at 24 months post-tracking was associated with being male (aOR = 0.61; 95%CI: 0.40–0.92) and initiating ART after test-and-treat recommendations (aOR = 0.16; 95%CI:0.10–

confidential in order to protect the privacy of participants. An excerpt of the qualitative data set is uploaded as supplementary information.

**Funding:** This study was made possible by the support of Wagner Foundation. Dr. Rosenthal's role was funded by the Israel Science Foundation - ISF (Grant 1073/18). The funders had no role in study design, data collection and analysis, decision to publish or preparation of the Manuscript.

**Competing interests:** I have read the journal's policy and the following authors of this manuscript have the following competing interests: Anat Rosenthal is an editorial board member of PLOS Global Public Health. The other Authors have no other conflicts of interests.

0.25). The qualitative analysis revealed that clarity of the visit's purpose, TRACE's caring approach changed patient's mindset, enhanced sense of responsibility and motivated patients to resume care. We recommend integrating tracking programs in HIV care as it led to increase patient follow up and patient behavior change.

## Introduction

Human immunodeficiency virus (HIV) is still a significant challenge globally, with 38.6 million people living with HIV (PLHIV) at the end of 2021 [1]. The rapid scale-up of universal antiretroviral therapy (ART) coverage–"test-and-treat"–since 2016, strategizing early disease detection coupled with early ART initiation regardless of disease stage, has increased ART coverage to 35 million PLHIV worldwide [2]. Antiretroviral therapy coverage worldwide increased from 25% in 2010 to 75% in 2021 [3]. This strategy has enabled health systems to improve the mortality, morbidity, and overall quality of life of millions of PLHIV [4]. The success of its implementation has allowed PLHIV to live a longer and more productive life and has made HIV a manageable and chronic disease [5]. Preventing clients on ART from LTFU is vital for preserving the progress made toward ending the HIV pandemic by 2030 [6–9].

LTFU is defined as patients enrolled in ART programs who have missed scheduled clinical visits or drug collection appointments and have been out of care for an estimated 180 days or more from their last visit. Some countries and regions have adopted shorter periods, with the shortest being 60 days [10, 11]. Regardless of the defined period adopted, LTFU threatens the success of HIV care programs. In Sub-Saharan Africa, 56% of HIV program attrition is attributed to LTFU [12] and is prevalent among men, women, and children, as well as other specific groups living with HIV [13–15]. In Western Kenya, the incidence of LTFU was 28.1% and 23.8 per 100 person-year in men and women between 2001 and 2007 [13]. Leshargie and al, in data published between 2005 and 2020 that the trend of LTFU in adolescents living with HIV and initiated into treatment in Sub-Saharan Africa was increasing over time and older adolescents (15–19 years old) 43% were more likely to be LTFU than the younger ones (10–14 years old) [16]. LTFU has been reported to double the mortality risk for HIV-infected children who started treatment in South Africa [17]. Furthermore, even if patients return to care after being LTFU, they have been reported to be more likely to die than patients who attended the clinic regularly [18]. With irregular treatment and follow-up, patients who have left care have been reported to be a significant driver for new cases of HIV infections [19].

In Sub-Saharan Africa reasons for LTFU range from individual (psychological) to socio-economic and programmatic. The current literature describes two levels of barrier to retention leading to LTFU. A first level of barriers that are more socio-economic programmatic and environmental such as financial difficulties (transportation costs, lack of food, inability to pay service fees), mismatched scheduling, unexpected events, distances to health facilities, which are found to be associated with high rate of return to care [20–24]. A second level of barriers are more medical and psychological such as side effects, forgetfulness, fear of stigma, lack of motivation, denial, personal beliefs, treatment fatigue, which requires focused inter-personal interventions [21, 25, 26]. To address these barriers, the World Health Organization (WHO) has recommended that HIV care programs implement clinical and community-based interventions to support ART adherence and patient RiC [27, 28]. The current body of literature reports that interventions targeting both level of barriers, have been proven cost-effective in preventing LTFU [29–33]. Programmatic interventions like transport reimbursements, direct

cash transfer, newer drug formulation, differentiated health care service delivery have been implemented in Sub-Saharan Africa to address socio-economic and medical causes of LTFU [34, 35]. In addition, interpersonal intervention examples for LTFU patients including phone calls, text messages, peer counsellors, patient navigators, or lay counsellors to trace LTFU patients were proven to be successful interventions in addressing medical and psychological barriers to retention [30, 32, 34, 36–41].

In the past two decades, Malawi–one of the Sub-Saharan countries with the highest prevalence of HIV–has made great progress in the fight against the HIV pandemic [42]. With various interventions and with a public health approach, including an early "test-and-treat" policy in 2016, the prevalence of HIV in the past 20 years has fallen from 15.5% to 7.7% [43, 44]. Antiretroviral therapy coverage and viral load suppression were 91% and 87.3%, respectively, in 2021 [42, 45]. However, LTFU in the era of test-and-treat remains a challenge for the HIV national program [46–48]. Currently, the Malawi national guidelines assign LTFU status to all patients missing appointments for drug refills after two months (eight weeks) [49]. As patients engage and disengage throughout the continuum of care, LTFU constitutes a real threat to test-and-treat success in Malawi.

Recognizing the detrimental effect of LTFU on HIV programs, we created a preventive default tracking strategy named the Tracking Retention And Care Enrollment (TRACE) program in 2016 in the rural Neno District of southeastern Malawi. TRACE program capitalized on high correlation of attitudes and subjective norms to behavioral intent and behavior change [50]. As a community-based intervention, TRACE program invites patients to return to care. TRACE is preventive and focuses on the linkage between the health facility and the patient by providing patient-centered support. By aiming to prevent LTFU, TRACE improves retention, with the aim to improve viral load suppression and overall patient outcomes. However, the program has not been described and further knowledge of the TRACE program will serve to inform design and implementation of community interventions that aim to prevent LTFU and increase patient re-engagement. Therefore, by using a mixed-methods approach, we aimed to present a description of patient re-engagement, RiC, and patient characteristics after being tracked by the TRACE program along with program acceptability from the perspectives of the beneficiaries.

## Methods

### Study site

The Neno District is a relatively newly designated district, separated from the Mwanza District in 2003 under the national decentralization program, with a poorly developed health infrastructure and limited access to healthcare services. Healthcare services in the district include a district hospital, a smaller community hospital, and 12 healthcare centers. Due to the poor infrastructure in the district, the district hospital, community hospital, and 10 out of 12 healthcare centers cannot be accessed via a paved road. The population in the district is estimated to be 165,000, of which more than half live in extreme poverty with little or no access to clean water or electricity [51, 52].

### Neno's model of care delivery

Partners In Health, locally known as Abwenzi Pa Za Umoyo (APZU), has accompanied and supported the Malawian healthcare system since 2007 to promote the delivery of high-quality healthcare to the most vulnerable and marginalized populations and has a strong bond with the Neno District health officials. The core of APZU's model lies in a robust community healthcare network and an Integrated Chronic Care Clinic (IC3) model [53, 54]. The

community health program aims to bridge the gap between the healthcare facilities and the community. It is operationalized by a network of nearly 1,300 CHWs volunteers who live in the community they serve. CHWs receive training, ongoing education, service delivery tools and a monthly volunteer stipend (38USD). CHWs completement and alleviate the burden on Health Surveillance Assistant, ministry 's of health cadres that are often overworked within the health system in Malawi [55, 56]. Each CHW regularly visits 20 to 30 households monthly and provides an initial screening assessment for conditions such as HIV, malnutrition, maternal and child health, mental health, noncommunicable diseases (NCDs), basic hygiene, and health education [57]. Depending on the outcome of the screenings, household members are referred to health facilities for diagnostics and management. When CHWs visit the households, they provide treatment support to patients already enrolled in various care programs at the facilities.

In the health facilities, the integrated chronic care clinic aims to provide free of charge and decentralized high-quality care to patients diagnosed with and treated for conditions such as HIV, tuberculosis, palliative care, malnutrition, maternal and child health, mental health, and NCDs. The integrated chronic care platform is designed as a hub-and-spoke model, where the "hubs" are the two hospitals in the district which exist as the referral centers for chronic care patients. These "hubs" can provide additional staff, supplies, and medications as needed to support the primary health center's IC3 clinics–the "spokes" in provision of care weekly or fortnightly depending on the number of patients.

In this model, PLHIV and NCD patients are seen in the same clinic with the same providers without differentiating between them to reduce stigma and assure "one-stop-shop" efficiency [53, 58, 59]. Upon each visit patients received daily group or individual health related counselling. Since 2007, APZU and the Ministry of Health in the Neno District have been implementing an electronic medical record (EMR) system for patient-level data management, which is used in conjunction with the national paper-based master card system to monitor visits, patient flow, and missed appointments [60]. Upon enrollment into IC3, patients are asked to consent to receive in-person or telephone call reminders if they miss appointments. All patients have a paper master card that captures demographics, visit details and medication data, which is entered into an EMR system after each IC3 clinic day. The IC3 model of care was shown to increase ART enrollment, decrease travel distance, and improve retention in care [61]. As of July 1st, 2021, the HIV cohort receiving ART in Neno through the IC3 model comprised 8291 people.

## TRACE strategy

The TRACE program is a community-based program woven into the care package provided to chronic care and HIV patients to prevent LTFU. TRACE program uses CHWs and dedicated non-clinical staff termed "TRACE trackers" to reach out to any patients enrolled in the integrated chronic care clinic who have missed their scheduled appointment or medication refill for 2 weeks following the appointment date. Per the Malawi national HIV guidelines, when HIV-positive patients miss their scheduled appointments and do not return to the clinic after eight weeks, they are defined as LTFU or "defaulters" [62]. In keeping with the Neno District TRACE strategy, patient tracking for retention was initiated at 2, 4, and 6 weeks (of a missed appointment) rather than waiting the full eight weeks, defined by the national guidelines, to prevent LTFU and patient defaulting (Fig 1).

TRACE trackers are lay non-clinical staff tasked to focus on preventing LTFU by tracking patients and providing a more holistic assessment of the patient's determinants of LTFU. TRACE trackers receive psychosocial training, HIV testing and counselling training, ongoing

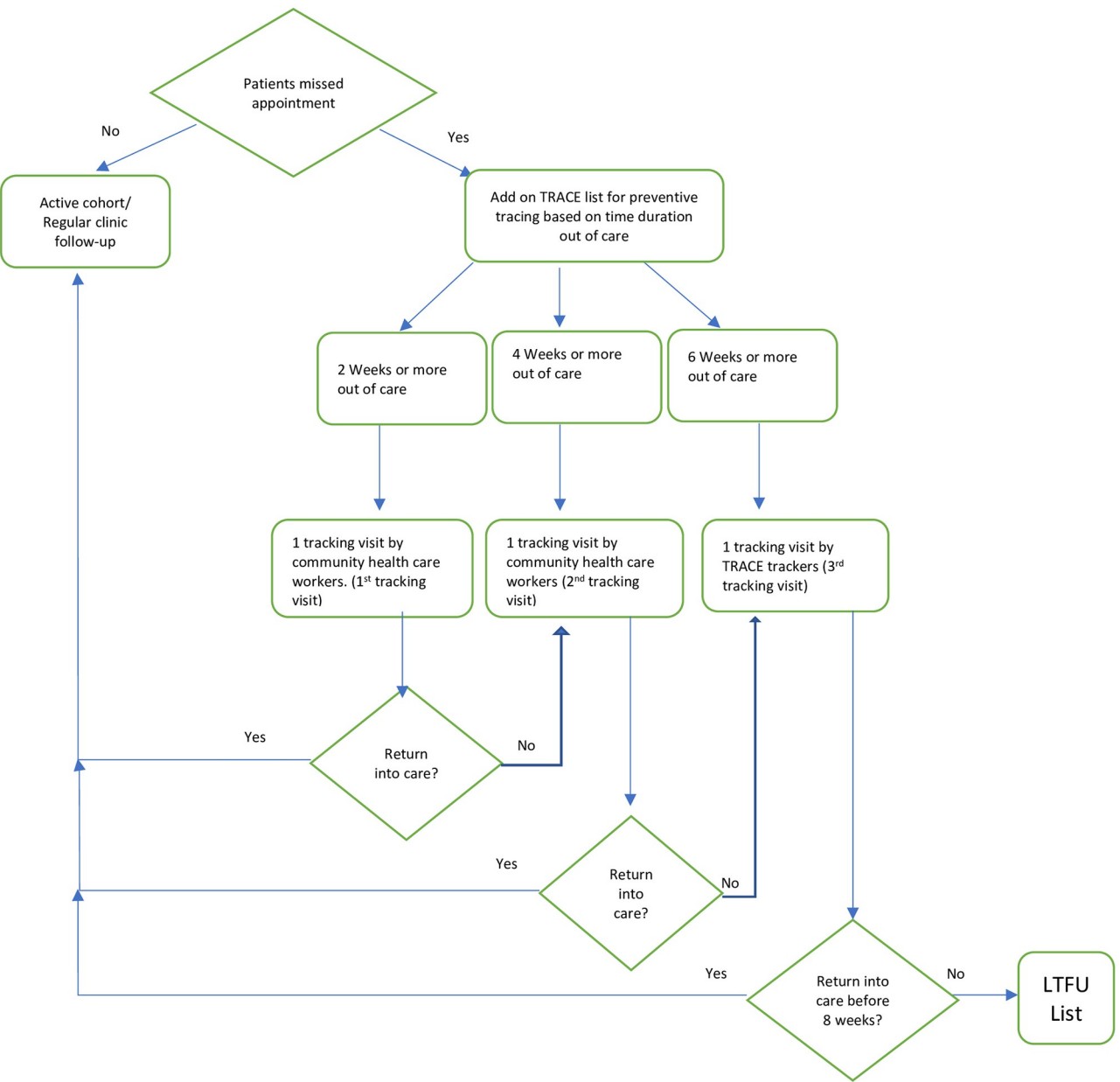

**Fig 1. Presents the algorithm for tracking eligible patients during monthly tracking activities.** This process is automated within the electronic medical system that generates 3 lists of patients to be tracked based on the duration of time being out of care.

education on HIV patients' management. Trackers are by nature a component of the community program in Neno but they are supervised by a clinical manager. TRACE uses guidelines and service delivery management tools to assess patients' needs and to determine appropriate program for reference.

TRACE program is continuous and woven into the care package provided to chronic care and HIV patients to prevent LTFU. As the tracking is continuous, twice a month a tracking list of patients who have missed appointments for 2, 4, or 6 weeks or more is generated using EMR. The tracking list is sent to the CHW team to ensure the tracking of missed appointments

                          

of under 6 weeks, whereas the TRACE trackers track patients who have missed appointments for six weeks or longer. TRACE relies on CHW because of their complete integrations within communities, health facilities and district health programs. During the visit, the CHWs remind patients of their missed appointment and inform them to return to the facility for a visit, which will be updated in the EMR system. In addition, CHWs collect information and refer (sometime accompany) patients to the clinic for linkage to care. Therefore, patients on the TRACE trackers list already had 2 short visits from the CHW team which constitute early reminder. Suppose the patient cannot be found or does not return to the facility for >6 weeks. In that case, the TRACE tracker is now sent to the patients' homes to remind them of their appointment, assess why they missed the appointment. During this third visit the patient receive a deeper psycho-social assessment by TRACE tracker. The length of this visit varies from few minutes to half an hour accordingly to nature of the reason for missing schedule appointments. After the tracker introduce her/himself, states the purpose of the visit, time is given to patients to share their concerns and/or challenges and to discuss possible solutions with the tracer. For each issue identified the patient provide their understanding of the issue at hand and possible solutions. At the end patients are referred to appropriate APZU programs like clinical programs, psycho-social, financial assistance service, and nutrition services to address possible root causes for missing appointments. Reference to appropriate services is done first verbally referral with patient in addition to electronic referral (with patient's details) is sent to the receiving programs.

## Study design

A mixed-methods approach combining quantitative and qualitative investigative methods was used to describe the characteristics of patients who return to care after being tracked by TRACE and patients' perspectives toward the TRACE program. A sequential explanatory design was used with two successive phases, quantitative and qualitative. The quantitative outcomes were defined as 1) Re-engagement or return to care of patients out of care beyond 6 weeks of a missed visit 2) RiC at 6-,12- and 24-months post tracking and 3) descriptive characteristics of these patients as factors associated with their retention in care post tracking. The quantitative phase was conducted first and followed by the qualitative phase to illustrate the results obtained in the quantitative phase. Based on the results of the quantitative phase, an interview guide was developed, and 26 participants (2 participants per facility) were purposefully selected for their specific characteristics (facility, 3 or more tracking visits, re-engaged with care after tracing) for semi-structured interviews [63]. The qualitative results provide a deep understanding of why patients who were preventively tracked re-engaged in care in Neno. In the discussion of the outcomes of the entire evaluation study, results from both quantitative and qualitative data were integrated together. Qualitative results were used to explain the quantitative findings and to formulate recommendations [64].

## Target population

This evaluation study included all HIV-positive patients in the Neno District IC3 HIV cohort who were tracked between January 2018 and June 2019. They had been tracked because they had missed their appointments for a clinical review or for a drug refill for 6 weeks or more.

## Quantitative phase

**Quantitative data collection.** De-identified data (S2 and S3 Data) was collected using TRACE outcome monthly EMR reports and ART registers with 1100 patients tracked between January 2018 to June 2019. Demographic variables were extracted from the EMR system,

including gender, age, and address, with clinical variables including the year of ART initiation, CD4 at initiation, ART education session at enrollment, tuberculosis (TB) status at registration (TB positive, TB negative, TB suspected), WHO clinical stage at enrollment (WHO I, II, III or IV) [65], body mass index (BMI), and the enrollment status of the patient (active in treatment, defaulted, transferred out, died). Data were cleaned, and any missing data were completed using ART registers at the facilities.

## Data analysis

STATA version 14 was used for descriptive statistics. Variables reported were age, gender, year of ART initiation, BMI, CD4 at initiation, the reason for ART start, ART education, TB status at registration, and ART initiating facility. We computed descriptive statistics such as frequency distribution, and the percentages for all independent and dependent variables. To describe the level of variation in dependent variable retention by various demographic, clinical, and facility characteristics, we used Chi-square (Table 1). Results for continuous variables, including age and year on ART, were presented by the median and interquartile range (IQR). Meanwhile, categorical variables such as age group, sex, BMI, ART start after test-and-treat era, CD4 at ART initiation, clinical staging at ART initiation, ART education session done (or not), and TB status at registration were presented as a proportion of the variables of the 1100 patients listed. The primary analytic approach was the intention to treat, where participants were analyzed in the category to which they were assigned, regardless of whether or how often they were seen by CHWs or TRACE field workers.

The primary study outcome was retention after returning to care. Retention was measured from the day of re-engagement (i.e., the origin). A patient was to be considered retained at any given time if she/he at the time of analysis (6-, 12-, 24-months) was alive and enrolled in ART care or transferred to another HIV program after re-engagement in care. A patient was considered *not* retained if she/he defaulted, stopped ART, or died after re-engagement in care and did not reach the time of analysis (6-, 12-, 24-months). When estimating retention (to compare the proportion retained after successful tracing), we used generalized estimating equations with a log link and binomial distribution to estimate the risk ratio at 95% confidence intervals. We used an exchangeable correlation matrix and a robust variance estimator to account for correlation. Multivariate logistic regression was also used to identify patient characteristics independently associated with the outcome of return to care after TRACE in the study population. According to Malawian clinical guidelines, default was defined as patients not returning to the clinic for at least eight weeks after their scheduled visit (the date on which the last ARV refills were expected to be finished).

## Qualitative phase

**Qualitative data collection.** From the pool of participants, 25 were interviewed instead of the 26 who had been scheduled for interviews, as one had withdrawn from the study after being contacted without providing an explanation. Recruitment and interviews were conducted between August to December 2021. Twenty-five in-depth semi-structured interviews were conducted with ART active patients purposefully selected (facility, at least 3 tracking visits, re-engaged in care after tracing) from the list of patients tracked preventively between January 2018 and June 2019 [63, 66]. Selection was based also on retention after successful tracking by the TRACE team and willingness to participate in the study. Only patients 18 years of age or older, who could give consent, were included in this study. The interviews were conducted in the Chichewa language, recorded, and transcribed. The interviews aimed to capture

**Table 1. Baseline characteristics of study participants.**

| Descriptive and Clinical Characteristics | Study Participants n(%) = 1100 |
|---|---|
| Age (median, IQR) | 30 (23–38) |
| **Age group** | |
| <15 | 91(8%) |
| 15–29 | 435(40%) |
| 30–39 | 347(32%) |
| 40–49 | 156(14%) |
| >50 | 71(6%) |
| **Sex** | |
| Female participants | 630 (57%) |
| Male participants | 470 (43%) |
| **Year of ART start (median; range)** | 2016 (2012–2018) |
| **BMI categories** | |
| Below 18.5 | 230(21%) |
| 18.5–24.9 | 762(69%) |
| 25.0–29.9 | 87(8%) |
| 30.0 and Above | 16(2%) |
| **ART free of charge at start (after June 2016)** | 513 (64%) |
| **CD4 at start of ART (median, IQR)*** | 256 (170–346) |
| <200 | 76(30%) |
| 200–500 | 167(66%) |
| >500 | 8(3%) |
| **WHO Clinical Stage ART start Number (%)** | |
| WHO stage I | 646 (49%) |
| WHO stage II | 92 (29%) |
| WHO stage III | 188 (29%) |
| WHO stage IV | 27 (34%) |
| Test & Treat | 147 (13%) |
| **Number (%) ART education delivered** | |
| Yes | 999 (91%) |
| No | 4 (1%) |
| Not recorded | 97 (8%) |
| **Number (%) TB status at registration** | |
| Confirmed TB on treatment | 20 (42%) |
| TB treatment complete | 23(34%) |
| TB suspected | 4(27%) |
| Unknown | 1362(56%) |

the tracking elements from the patients' perspectives and the impact of the tracking on patient retention and re-engagement in care.

## Data analysis

Transcripts of the interviews conducted in Chichewa were translated into English (S1 Data) and coded using Atlas (version 14). The coded data were summarized and analyzed using content analysis, a count-based approach to determine the frequency of themes and categories in a qualitative data set [63, 67]. Emerging themes and categories were discussed by the study team. Themes and categories were based on the study goals of deepening our understanding of the

tracking, patient characteristics, and impact of the program from a patient perspective. The analysis focused on patients' opinions and perspectives on the impact and usefulness of TRACE.

### Ethics

All participants interviewed in the qualitative phase of this study gave written consent or oral consent if they could not read, in which case the Chichewa version of consent was read to them, and oral consent was given. Approval from the Neno ethics and research committee was obtained to access secondary data from EMR, TRACE outcome reports, and ART registers. The institutional review board of the National Health Science and Research Committee Malawi approved this study registered under Protocol # 21/07/2731.

## Results

### Quantitative results

**Characteristics of study participants.**   Between January 2018 to June 2019, a total of 1100 individuals had a TRACE visit, and of these, 1028 patients met the inclusion criteria of physical patient verification at a facility on master cards and registers. We excluded 72 participants from the primary outcomes as they had started and transferred out on a day of registration when such information was missing on EMR records (Fig 2).

The median age was 30 years (IQR: 23 years–38 years), with most study participants being adults (86%), and female participants accounted for 57%. Approximately 49% of patients started treatment at WHO clinical stage I before 2016 and the universal test-and-treat era. Overall, 251 study patients (23%) had at least a baseline CD4 count recorded at study entry with a median baseline CD4 cell count of 256 (range: 170–346/mm3) (Table 1).

**Return to care.**   Out of the 1028 patients eligible for the study with a 6-week missed appointment, 982 patients (96%) returned to care for at least one follow-up visit after tracking (Fig 3). However, no difference in return proportion by gender and age (p-value >0.05) (Tables 2 and 3).

**Retention in care.**   We reviewed retention at 3 points in time (6-,12-,24- months) post tracking, 906 (88%) were retained in care at 6 months, 864 (84%) at 12 months, and 839 (82%) at 24 months (Table 4).

We used logistic regression with univariate and multivariate analyses to examine potential demographic and clinical factors associated with re-engagement in care at 6 months post-TRACE (Table 5). According to the univariate and multivariate model, having WHO stage IV and initiating ART during the test-and-treat era were associated with an increased likelihood of retention-in-care. The multivariate model was constructed by including variables with a p-value<0.2 in the univariate analysis and further adjusting by age and gender. Factors associated with retention in care were people at WHO stage IV (aOR 0.18; 95% CI 0.06, 0.54) versus people at WHO stages I-III, and those who started ART during the test-and-treat era (in 2016 and later) (aOR 0.08; 95%CI 0.06, 0.18) versus those who started before the test-and-treat era (before 2016). Gender and BMI were not associated with RiC.

In the univariate and multivariable analyses, we examined factors associated with retention-in-care at 12 months post-TRACE (Table 6). According to the univariate and multivariate models, young people between 15 and 29 were more likely to be retained in care (aOR 0.18, 95% CI 0.03, 0.88, p-value <0.05 for both). Starting treatment at WHO stage IV was not associated with RiC. Starting ART after 2016 was highly associated with RiC (aOR 0.08, 95% CI 0.04–0.16, p-value <0.001). Gender and BMI were not associated with re-engagement in care.

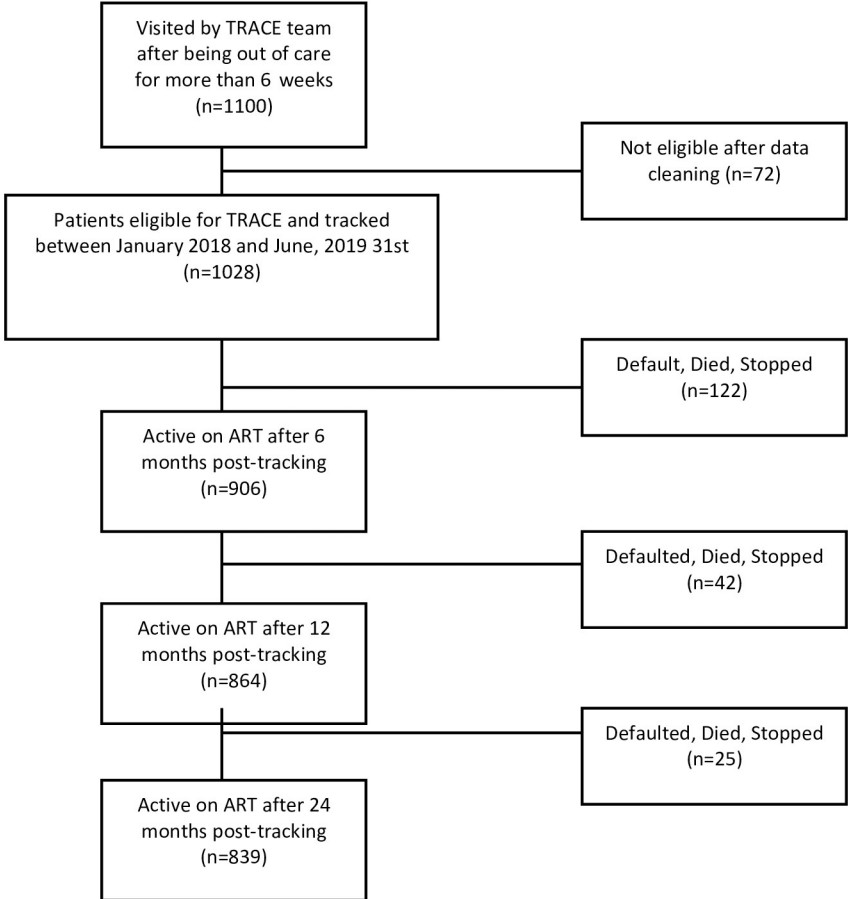

**Fig 2. Describes breakdown of all patients tracked between January 2018 to June 2019 by TRACE team reported on the 6-week list and their programmatic outcomes at 6,12,24 months.**

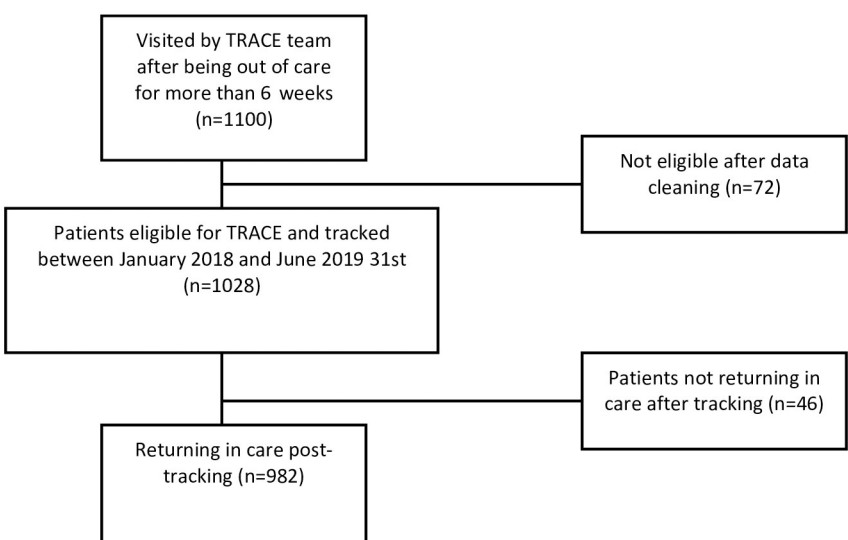

**Fig 3. Describes gross proportion of patients tracked between January 2018 to June 2019 by TRACE team reported on the 6-week list returning and not returning to care.**

**Table 2. Return to care after being tracked.**

| Patients tracked | Returned to Care | | |
|---|---|---|---|
| | **No** | **Yes** | **Total** |
| Female | 29 (5%) | 549 (95%) | 578 |
| Male | 17 (4%) | 433 (96%) | 450 |
| Total | 46 (4%) | 982 (96%) | 1,028 |

**Table 3. Return to care.**

| Patients tracked | Returned to Care | | |
|---|---|---|---|
| | **No** | **Yes** | **Total** |
| <15 years old | 2 (2.35%) | 83 (97.65%) | 85 |
| 15–29 | 19 (4.79%) | 378 (95.21%) | 397 |
| 30–39 | 12 (3.63%) | 319 (96.37%) | 331 |
| 40–49 | 7 (4.79%) | 139 (95.21%) | 146 |
| >50 | 6 (8.70%) | 63 (91.30%) | 69 |
| Total | 46 (4.47%) | 982 (95.53%) | 1028 |

**Table 4. Follow-up outcome by time point among TRACE clients.**

| Post-tracing outcome Milestone | Number of clients tracked | Alive on ART | Defaulted | Died | Stopped ART | Transferred out | RiC (Alive +Transferred out) |
|---|---|---|---|---|---|---|---|
| **6 months** | 1028 | 859 | 118 | 3 | 1 | 47 | 906 (88%) |
| **12 months** | 1028 | 808 | 158 | 5 | 1 | 56 | 864 (84%) |
| **24 months** | 1028 | 754 | 175 | 10 | 4 | 85 | 839 (82%) |

**Table 5. Factors associated with 6-month return to care and mortality outcomes among HIV-infected male patients on antiretroviral therapy (ART).**

| Variable | Sub-category | OR (95% CI) | *P*- value | AOR (95% CI) | *P*-value |
|---|---|---|---|---|---|
| Age | <15 | Ref | | Ref | |
| | 15–29 | 0.38 (0.15–0.98) | 0.05 | 0.30 (0.06–1.62) | 0.16 |
| | 30–39 | 0.47 (0.18–1.23) | 0.12 | 0.27 (0.05–1.44) | 0.13 |
| | 40–49 | 0.59 (0.20–1.70) | 0.32 | 0.66 (0.11–4.04) | 0.65 |
| | >50 | 0.48 (0.15–1,53) | 0.21 | 0.41(0.06–2.74) | 0.36 |
| Gender | Female | Ref | | Ref | |
| | Male | 0.75 (0.52–1.10) | 0.14 | 0.85 (0.50–1.47) | 0.57 |
| Baseline WHO Stage | I | Ref | | 1 | |
| | II | 1.23 (0.57–2.66) | 0.60 | 0.65 (0.29–1.50) | 0.31 |
| | III | 1.16 (0.66–2.05) | 0.59 | 0.67 (0.34–1.31) | 0.24 |
| | **IV** | **0.33 (0.14–0.84)** | **0.02** | **0.18 (0.06–0.54)** | **0.002** |
| BMI | Below 18.5 | Ref | | Ref | |
| | 18.5–24.9 | 0.73 (0.43–1.26) | 0.26 | 1.17 (0.59–2.33) | 0.64 |
| | 25.0–29.9 | 0.68 (0.29–1.59) | 0.37 | 1.10 (0.37–3.33) | 0.86 |
| | 30.0 and above | 1.38 (0.17–11.09) | 0.76 | 1.76 (0.19–15.70) | 0.61 |
| Initiation period | <2016 | Ref | | Ref | |
| | **≥2016** | **0.10 (0.06–0.17)** | **<0.001** | **0.08 (0.06–0.18)** | **<0.001** |

**Table 6. Factors associated with 12-month return to care and mortality outcomes among HIV-infected male patients on antiretroviral therapy (ART).**

| Variable | Sub-category | OR (95% CI) | P- value | AOR (95% CI) | P-value |
|---|---|---|---|---|---|
| Age | <15 | Ref | | Ref | |
| | **15–29** | **0.29 (0.12–0.70)** | **0.01** | **0.18 (0.03–0.88)** | **0.04** |
| | 30–39 | 0.49 (0.20–1.20) | 0.12 | 0.27 (0.05–1.44) | 0.21 |
| | 40–49 | 0.42 (0.16–1.10) | 0.08 | 0.66 (0.11–4.04) | 0.13 |
| | >50 | 0.40 (0.14–1,14) | 0.09 | 0.41(0.06–2.74) | 0.26 |
| Gender | Female | Ref | | Ref | |
| | Male | 0.83 (0.60–1.17) | 0.30 | 0.85 (0.51–1.34) | 0.45 |
| Baseline WHO Stage | I | Ref | | Ref | |
| | II | 1.31 (0.67–2.55) | 0.43 | 0.43 (0.17–1.05) | 0.06 |
| | III | 1.56 (0.93–2.64) | 0.20 | 0.59 (0.27–1.27) | 0.18 |
| | **IV** | **0.51 (0.21–1.25)** | **0.14** | **0.12 (0.04–0.16)** | **0.002** |
| BMI | Below 18.5 | Ref | | Ref | |
| | 18.5–24.9 | 0.73 (0.43–1.26) | 0.42 | 1.17 (0.59–2.33) | 0.64 |
| | 25.0–29.9 | 0.68 (0.29–1.59) | 0.89 | 1.10 (0.37–3.33) | 0.86 |
| | 30.0 and above | 1.38 (0.21–4.72) | 0.99 | 1.76 (0.19–15.70) | 0.61 |
| Initiation period | <2016 | Ref | | Ref | |
| | **≥2016** | **0.11 (0.08–0.18)** | **<0.001** | **0.08 (0.04–0.16)** | **<0.001** |

We also examined factors associated with RiC at 24 months post-TRACE in the univariate and multivariate analyses (Table 7). According to the univariate and multivariate model, being male (aOR 0.61; 95%CI 0.40–0.92) and starting ART in 2016 or later (aOR 0.16; 95%CI 0.10–0.25) were associated with an increased likelihood of RiC as compared to older people, WHO stages I-III, and starting ART between 2007–2015, respectively. Age and BMI were not associated with RiC at 24 months.

**Table 7. Factors associated with 24-month return to care and mortality outcomes among HIV-infected male patients on antiretroviral therapy (ART).**

| Variable | Sub-category | OR (95% CI) | P- value | AOR (95% CI) | P-value |
|---|---|---|---|---|---|
| Age | <15 | Ref | | Ref | |
| | 15–29 | 0.27(0.11–0.64) | 0.01 | 0.32(0.09–1.17) | 0.09 |
| | 30–39 | 0.38(0.16–0.91) | 0.03 | 0.44(0.12–1.63) | 0.22 |
| | 40–49 | 0.37(0.14–0.94) | 0.04 | 0.57(0.14–2.29) | 0.43 |
| | >50 | 0.25(0.09–0.68) | 0.01 | 0.35(0.08–1.50) | 0.16 |
| Gender | Female | Ref | | Ref | |
| | **Male** | **0.63 (0.46–0.87)** | **0.01** | **0.61 (0.40–0.92)** | **0.02** |
| Baseline WHO Stage | I | Ref | | Ref | |
| | II | 2.19(1.07–4.49) | 0.03 | 1.39 (0.61–3.16) | 0.44 |
| | III | 2.22(1.31–3.76) | 0.003 | 1.59 (0.81–3.15) | 0.18 |
| | IV | 0.84(0.33–2.13) | 0.72 | 0.50 (0.18–1.42) | 0.19 |
| BMI | Below 18.5 | Ref | | Ref | |
| | 18.5–24.9 | 0.73 (0.43–1.26) | 0.09 | 1.07(0.61–1.89) | 0.80 |
| | 25.0–29.9 | 0.68 (0.29–1.59) | 0.54 | 1.05(0.43–2.56) | 0.92 |
| | 30.0 and above | 1.38 (0.21–4.72) | 0.95 | 1.70(0.32–8.97) | 0.53 |
| Initiation period | <2016 | Ref | | Ref | |
| | **≥2016** | **0.13 (0.09–0.19)** | **<0.001** | **0.16(0.10–0.25)** | **<0.001** |

## Qualitative results

In the qualitative arm of the study, participants were men and women between the ages of 19 and 70. Inductive evaluation of the themes emerging from the qualitative interviews, participants revealed 4 key areas regarding the TRACE encounter, including 1) clarity in the purpose of the visit, 2) a caring and friendly approach, 3) motivating patients to return to care, and 4) ownership of their care and personal responsibility, which positively impacted their future outcomes.

**1) Clarity in the purpose of the tracking.**   During interviews, patients addressed the role of TRACE clearly, and the purpose of the tracking was to reconnect them to the clinic to prevent the dire consequences of non-adherence to ART. Patients participating in this study were able to state the clear connection between stable health and their tracking to bring them back into care. As explained by a participant in ART for more than 10 years:

*". . . the visit is to find out if our body are healthy, maybe we can be collecting medication without taking them, and they also see that the days are accumulating at the hospital, and they visit us to see if we are taking medication properly, to prevent sickness that comes in the future, and it can happen that they have lost their person so they encourage the person that everyone should be taking medication at the right time and live a healthy life.*

*Participant #17, woman, 43 years old.*

Within the same theme, a male farmer on ART for more than 5 years stated:

*"TRACE has helped a lot. In short, it has helped in saving my life by reminding me to go and get my drugs and also sometimes they ask you are ready to go to the hospital and get your medication".*

*Participant #25, man, 43 years old.*

The personal context of TRACE visits, in which a CHW or TRACE tracker comes to their house, provides patients with the support and guidance they need to resume care. This notion was stated by a grandmother who was on ART but was also responsible for looking after a young 16-year-old boy on ART:

*"Trace is a group of doctors that work with us patient who are discouraged the walk around the house and it's a good house, they remind us when we forget about taking medication and tell us to go, for me I know that they visit and encourage me when I am discouraged they come and find me and ask what has happened if I am sick, I don't have a guardian."*

*Participant #13, woman, 56 years old.*

As these quotes show, the explicit purpose of TRACE and its role in the system of care was clear to the clients. There was a clear understanding that TRACE was part of the IC3 clinic, as is shown in the next section.

**2) Caring approach of TRACE tracking.**   People living with HIV have long been victims of stigma. Stigma is often experienced as a lack of empathy from others and as being a victim of judgmental, harsh, and ugly behaviors. Patients described the TRACE tracking experience as opposed to the experience of the stigma they had often undergone previously. This approach was caring and friendly, making patients feel comfortable and open with trackers. Participants felt at ease in the presence of the trackers. Furthermore, the positive atmosphere and the

friendliness of the interaction with TRACE staff were experienced by clients as measures of success as the trackers went out of their way to support them:

"*Mmmh, I really stopped taking medication, what made me to stop taking medication is that people were talking a lot to me at this house when I touch the utensil they were saying I am going to infect them so I made a choice to stop taking medication so that I should die and for-get things and get sick that was when I saw friends that loved me followed me and made me start taking medication, right now I see that when people talk a lot about I should only care about my life.*

"*Participant #18, man, 30 years old.*

Another participant reiterated the same sentiment:

" *They wanted me to be a good person, I mean when they left their jobs and visited me they wanted me to be in good health, I should be like my friends.*

"*Participant #10, man, 22 years old.*

The friendly aspect of the relationship with TRACE team members was expressed by other participants as well:

"*They came to see me, with the way they came to see me as a friend not trace team and it when they told me to go to the hospital.*

"*Participant #11, man, 44 years old.*

Participants experienced the approach of the TRACE team as crucial to their re-engagement in the program. The framing of the relationship with TRACE team members as a friendly relationship played an important role in participants' willingness to reconnect with the clinic.

**3) Motivation to resume care.**   Patients miss appointments or default for multiple reasons, including lack of transportation, illness in the family, or travel, but regardless of the reason for the missed appointment, being physically far from the clinic affects the patient's mental moti-vation. A TRACE visit is a reminder to go back to the clinic and an opportunity for patients to receive encouragement and regain their motivation to reclaim the mental energy necessary to resume treatment. According to the interviewees, the TRACE visit helped support and facili-tate their willingness to return to care. Therefore, the encounter fulfilled the specific objective of motivating and encouraging patients by instilling the mental energy to resume treatment. These sentiments were shared by many of the participants:

"*I said, I was experiencing some nausea, just wanted to sleep and my feet were swollen, but the moment the people visited me. . . . I was energized and my energy got back to normal.*

"*Participant #18, man, 43 years old.*

Motivation to resume care was also addressed in the context of self-care:

" *. . .Motivate them to their health bodies and go back to clinic, I can just say it is to motivate them that their health is important, and they should take care of their body, the trace team motivates them.*

" *Participant #11, man, 44 years old.*

Another recurring theme was the visit's role in helping patients resume their medication routine, which had been neglected during the gap in care:

"...*Right now, [the TRACE visit] has impacted me and encouraged me to take my medication properly.*

"**Participant #13, woman, 56 years old.**

Being "back on track" was also seen as part of the patient's relationship with the clinic and the desire to reassume the role of patient:

"*It motivates us to see the date when to collect medication at the hospital at the right time, because of trace we are surprised and reminded to collect medication when they visit me today, I try my best not to be visited again to remind me, so it helps, helps to know time to collect medication is here.*

"*Participant #11, man, 44 years old.*

Tracking as a motivational tool worked as both a support mechanism and to invite people back into care.

**4) Ownership of care and personal responsibility.** Patients who neglected their appointments to the point of becoming eligible for 6-week tracking, often with a tracking visit, described awareness and understanding of their role as patients, and the importance of keeping appointments. During interviews, patients put forth multiple reasons for missing appointments, such as treatment fatigue, social and economic challenges, and comorbidities. To overcome these obstacles, an ownership mindset is required for patients to seek help or find new means for overcoming these obstacles. This step marks an internal process wherein patients' understanding change regarding adherence and level of responsibility. The understanding of patient responsibility in adherence to ART treatment were largely emphasized by participants as one of the major impacts of the tracking visit. The TRACE visit influenced patients' mindsets and intentions to pick up and take medications by helping patients see their role. Patients realized that they–and not only the clinic staff–bore responsibility for their fate. As a 32-year-old female participant, mother of 3 children, explained:

"... *intentionally I forgot, chose not to follow the date not to go on the right days and missing dates, but after being visited to advise me, I can see that things are going on well I am keeping the dates, I am not missing the taking medication and I am taking following the date.*"

*Participant #09, woman 32 year old.*,

"*The conversation was very fruitful, because when I came I started taking medication properly since I don't want them to come and remind me, so I follow what they told me and the doctors because the doctors take us in classes when we visit the hospital after being taught they tell us not to stop taking medication, it like when we stop we make this people repeat the same job several times when I am going to the hospital properly and taking medication collection this people doesn't have tough time with me, because they know that am following everything properly.*

"*Participant #12, man, 65 years old.*

The TRACE tracking process aids in reshaping the mindset and raising awareness about patient responsibility. When receiving a friendly, motivating, and purpose-oriented visit from

the TRACE trackers, patients attach meanings to the visit and change their mindset regarding adherence and connectedness to the clinic, as stated by one participant:

> ". . .I was not looking after my life, when they visited me and told me that they are from the hospital, I was very happy and in knew that I was playing too much and I changed I don't miss medication even when I am drunk according to what trace team told me, even when I am drunk, I take medication in the morning.
>
> " **Participant #13, woman, 56 years old.**

The role of the TRACE program in helping patients find the mindset that will enable them to start care again was seen by participants as crucial to their decision to reengage with care.

## Discussion

We observed that TRACE was associated with high rate of re-engagement, 96% of HIV clients with missed appointments returned to care after a TRACE visit. This high re-engagement rate shows how a community outreach strategy such as TRACE, even in rural and hard-to-reach areas in low- and middle-income countries, can contribute to address the adverse effects of LTFU and defaulting from care. Our results concur with similar tracking programs implemented in Sub-Saharan Africa, demonstrating the positive impact and high re-engagement rate of clients who disengaged from HIV care [68, 69]. Zhou et al., when analyzing data from the Asia-Pacific region, predicted that the risk of being permanently LTFU increases with the number of days out of care [70]. As such, TRACE helps mitigate the negative effects of LTFU.

Patients presenting as WHO stage IV during their ART enrollment were more likely (than stages I-III) to be active in care after 6- and 12 months post-tracking. The literature on retention post-tracking is scarce; therefore, the significance of this finding might be specific to patients in the Neno District. In contrast to this finding, a study on frequent disengagement and re-engagement in Kenya showed that patients at WHO stage IV were less likely to return to care after tracking [71]. Throughout the literature WHO stage IV at ART enrollment has been was shown to be an important characteristic associated with poor outcomes. For example, Mutasa-Apollo et al. found in a study in Zimbabwe that clinical stage IV was one of the factors associated with a high attrition rate [72]. Furthermore, a study conducted in Latin America showed that patients with WHO stage IV presented a high risk of dying in care even after ten years, regardless of events between enrollment and death [73].

Initiating ART during the test-and-treat era was found to be highly associated with being alive and active in ART at 6-, 12-, and 24-months post-tracking when compared to patients who initiated ART before the test-and-treat era. Initiating ART during the test-and-treat era was also associated with high RiC in other studies. Brown et al. reported 95.5% retention in a study conducted in both East and West Africa [74]. Patients enrolled during this period received ART care free of charge and mainly at an early stage of the disease, impacting their survival and RiC even with a history of missed appointments. Our results suggest that the early tracking of these patients could address disengagement among patients. This early tracking strategy could help maintain the gains achieved with the test-and-treat strategy. A meta-analysis of patient disengagement after initiation into ART showed that a tremendous falloff occurred between 6 and 12 months [12]. Udeagus et al. found that patients' continuous engagement increased in a group of patients after returning to care [75]. These findings show the importance of continuous efforts to identify and address patients' adherence to ART programs. Additional investigations are needed to establish facilitators of and barriers to sustaining retention.

At 24-month post-tracking, men in this study were more likely to be active in ART than women, which contrasts with findings from other studies on retention in male after-ART-care enrollment. Ochieng-Ooko et al., in Western Kenya for a 6-year study period of a large cohort of patients, also found that the LTFU rate was 28.1% per 100 person- year in men versus 23.8% in women after ART initiation [13]. These findings suggest that retention in care programs need to focus on women need. Finally, participants in the age group of 15–29 showed notable retention at 12 months post-tracking. Mackenzie et al. found the exact correlation between age group and retention rates when studying the effect of age on retention in Malawi [76]. Our results suggest that, when designing programs to mitigate LTFU, the needs of specific groups such as older patients (30-year-old and plus) must be taken into account to address these needs efficiently.

As the qualitative analysis shows, re-engagement in care in the Neno District relies heavily on the TRACE program. The success of the program could be attributed to several factors. First, the purpose of tracking to prevent the dire consequences of non-adherence to ART was made clear to participants. Second, the participants reflected on the caring approach of tracking teams, especially in cases where stigma had played a significant role in their previous retreat from care. Third, TRACE trackers motivated participants to continue with treatment regimens. Fourth, participants saw TRACE as crucial to better their understanding on their responsibility as patient and to remain in care. Ware and al, reported mechanisms initiated by the patients themselves like borrowing, saving and prioritizing when such responsibility is understood [77]. The human-centered approach behind TRACE was essential in successful re-engagement in care and positive outcomes. In other studies, Beach et al. found that a caring approach from healthcare workers made the provider-patient relationship more human (i.e., less formal), and it was associated with better adherence and better health outcomes among patients living with HIV [78]. In several studies, it was found that patients who missed appointments or defaulted were afraid of receiving harsh treatment from clinic staff; they needed a supportive, non-judgmental, and welcoming interaction with clinic staff as a first step or a "motivator" to enable their re-engagement in care [79–81]. The vital role played by positive patient-provider interactions in patient re-engagement found in our study is not unique. For example, Hurley et al. found that patient-provider communication could facilitate patient re-engagement and RiC [81]. In other studies, it was found that the interaction between the patient and the healthcare team greatly affected the re-engagement of patients after being out of care [78, 82].

The interaction between patients and trackers occurred outside the clinic environment, in patients' social spaces, such as patients' homes or nearby environments that support caring and thoughtful interactions [83]. The interactions between the TRACE team and the patients complement the care landscape in which the practice and experience of care take shape [84]. In the current study, patients and patient-provider interactions took center stage as core facilitators of re-engagement in care. That said, other components, such as proposing HIV self-testing, social support, and incentives, have also been found to be effective in patient re-engagement [80, 85, 86] Nevertheless, tracking that has a clear purpose, well explained to patients, informative, and that can convey a message of acceptance and friendship–as shown in our study results–is more likely to achieve re-engagement and positive outcomes. Tracking strategies have tremendous benefits for patient health and communities. As such, tracking should be intentionally programmed within HIV care programs, as it extends the landscape of care.

## Limitations

The current study had several limitations and indicated the need for further studies. First, it was conducted in one district implementing a combined NCD and HIV clinic which is not

common within the other districts in Malawi where there are separate clinics for these conditions. However, the patient tracking did not differentiate between types of conditions so we would expect the same findings if the TRACE strategy was implemented for any condition. Second, we did not explore other factors that could influence patients' re-engagement in HIV care, such as distance from health facilities, socioeconomic status, transportation, and geography [61, 87]. Third, the time between ART enrollment and patient outcome was not factored into our analysis, nor was the time between tracing and re-engagement. Fourth, a comparison group was not included in the analysis as all patients who missed appointments in the district were traced according to TRACE protocol. The sample used for the quantitative analysis was small, and this analysis was conducted in only one district. However, the mixed-methods approach provides a comprehensive understanding of patient re-engagement and allows us to explore long-term RiC post-tracking, thus adding to and supporting the knowledge we already have regarding retention and re-engagement in care. Further research should include the clinical outcomes of patients post-tracking with inclusion of control groups.

## Conclusion

This study describes the TRACE strategy as an effective way to reengage with chronic care patients who have missed an appointment. Most patients return to care through this strategy with high retention rates, especially among men, older individuals, and those enrolled at more advanced clinical stage, and enrolled under the universal test-and-treat guidelines. From the patients' perspectives, re-engagement in care resulted from the relationship with the clinic and TRACE's ability to create a safe return path. Based on the results of this study, we recommend that outreach strategies addressing LTFU be intentionally integrated into HIV program designs, with age-specific and gender-centered factors taken into consideration, to improve patient retention in lifelong ART.

## Supporting information

**S1 Checklist. Checklist of TRACE manuscript.**
(DOCX)

**S1 Data. Except of the qualitative data set.**
(DOCX)

**S2 Data. Quantitative data set of demographics.**
(XLSX)

**S3 Data. Quantitative data and outcome results.**
(XLSX)

## Acknowledgments

We are grateful to all patients at IC3 clinics in Neno. Special thanks to the Ministry of health team in Neno District for their support. Special thanks to M&E team, CHW team and TRACE team at APZU.

## Author Contributions

**Conceptualization:** Jean Christophe Dimitri Suffrin, Anat Rosenthal, Emilia Connolly.

**Data curation:** Moses Banda Aron.

**Formal analysis:** Jean Christophe Dimitri Suffrin, Anat Rosenthal, Clement Trapence, Moses Banda Aron.

**Investigation:** Jean Christophe Dimitri Suffrin, Linda Kamtsendero, Jonathan Kalua, Enoch Ndarama, Clement Trapence.

**Methodology:** Jean Christophe Dimitri Suffrin, Anat Rosenthal.

**Project administration:** Jean Christophe Dimitri Suffrin.

**Supervision:** Emilia Connolly, Luckson W. Dullie.

**Writing – original draft:** Jean Christophe Dimitri Suffrin.

**Writing – review & editing:** Jean Christophe Dimitri Suffrin, Anat Rosenthal, Linda Kamtsendero, Chiyembekezo Kachimanga, Fabien Munyaneza, Jonathan Kalua, Enoch Ndarama, Clement Trapence, Emilia Connolly, Luckson W. Dullie.

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
