## [Decision Letter · Decision Letter 0]

11 Jul 2023

PGPH-D-23-01074

Re-engagement and retention in HIV care after preventive default tracking in a cohort of HIV-infected patients in rural Malawi: A mixed-methods study

Dear Dr. SUFFRIN,

Thank you for submitting your manuscript to PLOS Global Public Health. After careful consideration, we feel that it has merit but does not fully meet PLOS Global Public Health’s publication criteria as it currently stands. Therefore, we invite you to submit a revised version of the manuscript that addresses the points raised during the review process.

We look forward to receiving your revised manuscript.

Kind regards,

Professor Toyin O. Togun

Academic Editor

Journal Requirements:

1. In the ethics statement in the Methods, you have specified that verbal consent was obtained. Please provide additional details regarding how this consent was documented and witnessed, and state whether this was approved by the IRB

2. Please send a completed 'Competing Interests' statement, including any COIs declared by your co-authors. If you have no competing interests to declare, please state "The authors have declared that no competing interests exist". Otherwise please declare all competing interests beginning with twhe statement "I have read the journal's policy and the authors of this manuscript have the following competing interests:"

3. In the online submission form, you indicated that "The data for this manuscript will be made available to the to the editorial team if need

be upon request through the corresponding author". All PLOS journals now require all data underlying the findings described in their manuscript to be freely available to other researchers, either 1. In a public repository, 2. Within the manuscript itself, or 3. Uploaded as supplementary information.

Additional Editor Comments (if provided):

Reviewers' comments:

Reviewer's Responses to Questions

**Comments to the Author**

1. Does this manuscript meet PLOS Global Public Health’s publication criteria? Is the manuscript technically sound, and do the data support the conclusions? The manuscript must describe methodologically and ethically rigorous research with conclusions that are appropriately drawn based on the data presented.

Reviewer #1: Partly

Reviewer #2: Yes

2. Has the statistical analysis been performed appropriately and rigorously?

Reviewer #1: Yes

Reviewer #2: I don't know

3. Have the authors made all data underlying the findings in their manuscript fully available (please refer to the Data Availability Statement at the start of the manuscript PDF file)?

Reviewer #1: Yes

Reviewer #2: No

4. Is the manuscript presented in an intelligible fashion and written in standard English?

Reviewer #1: No

Reviewer #2: Yes

5. Review Comments to the Author

Reviewer #1: Thank you for the opportunity to read and review this paper. Overall, I found the paper to be very interesting and relevant. Some sections will quite a lot of revision to improve the manuscript. Please find my comments below:

ABSTRACT

Overall, the abstract and key messages read very well. However, please note the following edit:

- What OR adjusted for? This isn’t clear

- The comma is misplaced between “patients ‘outcomes”… It should read “patients’ outcomes”

BACKGROUND

Very well written. However, please note the following edits:

- You have “by 20203” and should read “ by 2023”

- Once loss to follow up is abbreviated, it should read “LTFU” throughout the manuscript

- More detail is needed in describing the TRACE programme here. I can see that more detail is provided in methods section but I think a more detail is needed or atleast referenced here

METHODS

- Should read “via a paved road” not “by a paved road”

- Missing references for APZU information including the IC3 model

- How were participants purposively chosen for the interviews? By age/gender?? This is missing here

- Given that you participants younger than 18 years in the study, why was ethics not sought to facilitate consent/assent for them? Would have really interesting to hear their perspectives as well

RESULTS

- There are two Table 3s

- Quote on page 27 doesn’t really support the “caring approach” described. I would replace with more appropriate quote

- I am not sure I see nor understand the difference between “re-shaping mindset” and “motivation” here. Perhaps you could make this more clear.

DISCUSSION

I would say that this section needs the most revision. Please see suggested edits and comments below:

- I don’t think you can say TRACE is reason for re-engagement… It may be one the reasons as your results show but there could be other reasons as well e.g. other existing programmes in the area

- “At 6, 12, and 24 months, respectively, 88%, 84%, and 82% of patients traced were still alive and active on ART at the time of this analysis. Our study sheds light on RiC post-tracking at three points in time, representing a unique aspect of this study given that in most studies, only characteristics, facilitators, and barriers to re-engagement have been explored (53).” – Not sure what this adds to the discussion. I would remove.

- “Furthermore, a study conducted in Latin America showed that patients with WHO stage IV presented a high risk of dying in care even after ten years, regardless of events between enrollment and death (55). Despite this finding, a study on frequent disengagement and re-engagement in Kenya showed that patients at WHO stage IV were less likely to return to care after tracking. In the Sub-Saharan region, ART patients at WHO stages III and IV need more support to reach the ART care facilities (57). ” – This paragraph structure doesn’t make sense either. Paragraph should start with topic sentence and then supported by evidence in following sentences. Should be revised.

- “Second, initiating ART during…” – second to what?

- These findings call for gender-focused approaches, such as in male clinics, when addressing retention and re-engagement in care.” – I don’t find this sentence very strong at all… I would revise to say something about how your findings, unlike others, suggest that women also need to be focused on in interventions.

- “the needs of specific groups” – give examples based on your results

- “we did not explore other factors that could significantly influence patients’ re-engagement in HIV care, such as distance from health facilities, socioeconomic status, transportation, and geography” – why wasn’t this done?

Reviewer #2: This paper reports on an intervention (TRACE) designed to reduce loss-to-follow-up (LTFU) among patients attending the IC3 (integrated chronic conditions services) in Neno District, Malawi. LTFU is indeed a serious barrier to achieving HIV treatment goals (and presumably those for other diseases as well). The study presents results from a mixed method evaluation of TRACE as conducted over 2 years, but without any comparison site/ counterfactual. As such, it would benefit from being re-written as a process evaluation that examines some of the challenges, successes, opportunities and - crucially - mechanisms of action for how the intervention was implemented and experienced. At the moment, the paper remains at a fairly descriptive level and lacks conceptual and contextual data that would help readers understand the intervention better, and how it was theorised to reduce LTFU and whether these hypothesised pathways of change were achieved in reality. More specific feedback for each section is provided below.

INTRODUCTION - the authors are correct to identify LTFU as a significant challenge, and one that continues to undermine HIV treatment success in Malawi and elsewhere. However, despite referring to literature that has examined some of the reasons behind LTFU (eg ref #7) these reasons are not presented, nor how different models for improving retention in care/ reducing LTFU are theorised to address these reasons (text messages, peer counsellors, patient navigators etc). These interventions were presumably designed with different theories of change in mind, i.e. how they might address barriers at different levels such as psychosocial / individual motivation, fear of stigma/ disclousre, lack of transportation or travel distance, etc. This kind of background information would be relevant to presenting TRACE and its guiding principles so readers understand its specific rationale.

METHODS - Related to the above, more detail about the rationale for TRACE (and its "active ingredients") is required. Why were CHW selected rather than other forms of patient-centred support? What does patient-centred support actually include in the context of the IC3 programme? What were CHW tasked with doing, and which potential determinants of LTFU were they targeting specifically? There is very little detail here - are CHW volunteers or a paid cadre within the health system? Why were "trackers" added after 6 weeks following missed appointment? What do trackers do that CHW do not? What was the theorised pathway from adding the TRACE tracking to improved retention in care? (was it simply about remininding and motivating patients, or was there any kind of psychosocial mechanisms such as offering social support, increasing problem-solving skills, etc?)

Qualitative methods - how were interviewees purposively selected (ie based on what criteria and why?)

Quantitative methods - why were the specific variables selected for coomparison? Age and sex are fairly obvious (I think the researchers asked about sex rather than gender?) but why BMI and TB status?

RESULTS: For background context, how many (and what %) of individuals were NOT reached by the CHW or TRACE trackers, and why not? (not home, refused entry into the home). As the paper is focused just on those individuals who were successfully identified by TRACE trackers, it's hard to get a perspective on trackers' overall ability to find the identified individuals. We know 1100 were visited at home, how many did trackers try and fail to visit?

I found the division of findings across the 3 time periods very confusing, and not something that was addessed in Discussion. Perhaps there is a way to synthesise the time periods by describing temporal trend? For example, stressing that while certain types of patients are more likely to be retained over the shorter term, other categories of people may experience longer-term barriers. Also, do the socio-demographic profiles of those tracked by TRACE differ substantially from those who never missed appointments (or were missing for fewer than 6 weeks and thus were never eligible for TRACE)?

Page 23 - in the paragraph directly preceeding Table 6 there is a contradicton regarding whether or not age is associated with RiC at 24 months. Age and BMI are stated as not being associated but an earlier sentence includes "older people" as a category with lower RiC (compared to being male and starting ART post-2016).

Qualitative results - these offer the possibility of examining "mechanisms of action" of the TRACE intervention, but again, the lack of any context regarding how the programme was designed, the activities of the trackers and how these were theorised to change behaviour, make it difficult to know how the researchers identified their themes. The qualitative analysis reads like formative research, the kind of work that should have been conducted before desigining an intervention in order to idetify relevant barriers to RiC that the trackers could target. Given the focus on "motivation" and "reshaping mindset" in the results, I am assuming the researchers wanted the intervention to focus on individual-level determinants of LTFU. What about addressing some of the broader issues like transportation difficulties, fear of stigma and disclosure, isolation and poor social support etc? Is there a reason that these were not part of the trackers' focus and did any of these emerge from the data?

I wonder if findings could be re-structured to start with the qualitative results and link these to how the programme was supposed to create behaviour change, and then identify for which kinds of people (and through what processes) this appeared to happen (or happen more or less). Again, the structure and concepts from process evaluation (including any challenges of implementation) might be a useful framework for this paper.

DISCUSSION - the researchers highlight where there findings differ from that in the literature (e.g. younger and male patients displayed better RiC following tracking). Is there any qualitative data that helped explain this? The two data sets could have been better combined in this section, to give a holistic picture of what the barriers might be for the different socio-demographic groups and how the work of the trackers appeared to reduce these.

OVERALL: It is laudable that the intervention was able to maintain high rates of RiC / prevent further LTFU for patients who were visited by the TRACE programme, however, we still don't know whether this was a large proportion of those it aimed to contact in the first place, and there is little information for other programmes in terms of how the programme works, for whom, and in what circumstances so that the lessons learned can be feasibly replicated elsewhere.

6. PLOS authors have the option to publish the peer review history of their article (what does this mean?). If published, this will include your full peer review and any attached files.

**Do you want your identity to be public for this peer review?** For information about this choice, including consent withdrawal, please see our Privacy Policy.

Reviewer #1: No

Reviewer #2: **Yes: **Joanna Busza

---

## [Decision Letter · Decision Letter 1]

27 Oct 2023

PGPH-D-23-01074R1

Re-engagement and retention in HIV care after preventive default tracking in a cohort of HIV-infected patients in rural Malawi: A mixed-methods study

Dear Dr. Suffrin,

Thank you for submitting your manuscript to PLOS Global Public Health. After careful consideration, we feel that it has merit but does not fully meet PLOS Global Public Health’s publication criteria as it currently stands. Therefore, we invite you to submit a revised version of the manuscript that addresses the points raised during the review process.

We look forward to receiving your revised manuscript.

Kind regards,

Toyin O. Togun

Academic Editor

Journal Requirements:

Additional Editor Comments (if provided):

Reviewers' comments:

Reviewer's Responses to Questions

**Comments to the Author**

1. If the authors have adequately addressed your comments raised in a previous round of review and you feel that this manuscript is now acceptable for publication, you may indicate that here to bypass the “Comments to the Author” section, enter your conflict of interest statement in the “Confidential to Editor” section, and submit your "Accept" recommendation.

Reviewer #2: (No Response)

Reviewer #3: (No Response)

2. Does this manuscript meet PLOS Global Public Health’s publication criteria? Is the manuscript technically sound, and do the data support the conclusions? The manuscript must describe methodologically and ethically rigorous research with conclusions that are appropriately drawn based on the data presented.

Reviewer #2: Partly

Reviewer #3: Yes

3. Has the statistical analysis been performed appropriately and rigorously?

Reviewer #2: I don't know

Reviewer #3: I don't know

4. Have the authors made all data underlying the findings in their manuscript fully available (please refer to the Data Availability Statement at the start of the manuscript PDF file)?

Reviewer #2: Yes

Reviewer #3: Yes

5. Is the manuscript presented in an intelligible fashion and written in standard English?

Reviewer #2: Yes

Reviewer #3: Yes

6. Review Comments to the Author

Reviewer #2: I think the Introduction needs to be revised to make clear that this is not an evaluation of the TRACE programme (lines 124-125 imply that an evaluation is being reported). It is a descriptive study of the characteristics and experiences of participants over a specific period. It is not an outcome evaluation because there are no comparisons made to previous trends or other sites or populations, and it is not a process evaluation because here are no clearly identified process variables related ti implementation such as fidelity, feasibility, quality etc. In the Discussion, I suggest removing the hypothesised counterfactual on line 629 that including comparison to non-tracked patients would "... would have amplified the effect" - this implies that you are attributing the observed rates of retention in care to the intervention, which you absolutely cannot do with the data presented. There is an association, but you have no way of being sure that there were not exogenous factors at play (a coincidental media campaign, a different health intervention, a change in public health facility protocols or even just a random change in health seeking behaviour).

Given that the paper is descriptive - and, indeed, useful as an indicator of the type of proactive preventive approaches that can be taken with LTFU - I would still like to see a bit more detail on the way that the intervention approached determinents of LTFU. The paper mentions being clear in purpose and "person centred" but no details are given in terms of what the TRACE workers did when they say someone, how long a visit or call was normal, what kind of counselling or problem-solving was offered, the health promotion messages included etc. Was there a manual or guidelines to follow, or were TRACE staff trained in differentiated support? There is little here that someone keen to replicate the programme in a similar context would be able to adapt and try. This level of detail would also help contextualise the qualitative data on acceptability, so that readers could understand which components were most appreciated by LTFU patients, and what likely mechanisms of action for successful RiC were. I suggest expanding the TRACE Strategy section that starts on line 181.

Reviewer #3: PLOS Global Public Health

Re-engagement and retention in HIV care after preventive default tracking in a cohort of HIV-infected patients in rural Malawi: A mixed-methods study

Manuscript Number: PGPH-D-23-01074R1

Reviewer Report

Manuscript text Comment

LINE NUMBERS Next time please include line numbers as it makes it easier for the reviewer and you in correcting.

A back-to-care program was implemented in urban Lilongwe, Malawi; 85% were successfully traced, and reasons for patients defaulting from the HIV-care program were explored (37). Check sentence not sure there should be a ; there. Also, maybe briefly state the reasons for patients defaulting.

The integrated chronic care platform is designed as a hub-and-spoke model, where the two hospitals provide additional staff, supplies, and medications to support the primary health center “spokes,” which are visited weekly or fortnightly for IC3 clinics. What are spokes?

From the pool of participants, 25 were interviewed instead of the 26 who had been scheduled for interviews, as one had withdrawn from the study after being contacted. Please state reason for withdrawal if known.

Twenty-five in-depth semi structured interviews were conducted with ART active patients purposefully selected from the list of patients tracked preventively between January 2018 and June 2019. Define purposefully selected and provide a citation.

The coded data were summarized and analyzed using content analysis for emerging themes and categories discussed by the study team. Define content analysis and provide a citation.

All participants interviewed in the qualitative phase of this study gave written consent or oral consent if they could not read, in which case the consent was read to them, and oral consent was given. I assume it was because they could neither read nor write in English. Please be clear.

Participant #17, woman, 43 years old. Sometimes it is nice to give an additional indicator about the participant as it relates to the research question, for example, education or occupation.

Participant #25, man, 43 years old

" Participant #13, woman, 56 years old. Please move these lines down below the quotations to maintain the correct formatting.

NUMBER AND FORMAT OF QUOTATIONS Just a suggestion but you could instead write one paragraph of summarizing text and then put the relevant quotes in a table format to streamline as the quotes are making the results section longer.

With this intervention, 96% of HIV clients with missed appointments returned to care. I like that your discussion section starts by pointing out the strength of the study

Furthermore, a study conducted in Latin America showed that patients with WHO stage IV presented a high risk of dying in care even after ten years, regardless of events between enrollment and death (55). It seemed this paragraph should be merged with the one prior. Make sure to keep paragraphs with a similar idea/messaging formatted together.

Second, initiating ART during the test-and-treat era was found to be highly… Third, at 24-month post-tracking, men in this study were more likely to be active in ART… None of the previous paragraphs open up with First, …

Nevertheless, tracking that has a clear purpose and is well explained to patients, is informative, and can convey a message of acceptance and friendship – as shown in our study results – is more likely to achieve re-engagement and positive outcomes. Re-word this sentence as it is busy and not clearly communicating the intended message.

First, it was conducted in one district with very distinct characteristics. Not clear. What is unique about the district that makes it a limitation?

Third, the time between ART enrollment and patient outcome was not factored into our analysis, nor was the time between tracing and re-engagement. Why not?

7. PLOS authors have the option to publish the peer review history of their article (what does this mean?). If published, this will include your full peer review and any attached files.

**Do you want your identity to be public for this peer review?** For information about this choice, including consent withdrawal, please see our Privacy Policy.

Reviewer #2: **Yes: **Joanna Busza

Reviewer #3: **Yes: **Penda Johm

---

## [Editor Report · Decision Letter 2]

3 Jan 2024

Re-engagement and retention in HIV care after preventive default tracking in a cohort of HIV-infected patients in rural Malawi: A mixed-methods study

PGPH-D-23-01074R2

Dear Dr Suffrin,

We are pleased to inform you that your manuscript 'Re-engagement and retention in HIV care after preventive default tracking in a cohort of HIV-infected patients in rural Malawi: A mixed-methods study' has been provisionally accepted for publication in PLOS Global Public Health.

Best regards,

Toyin O. Togun

Academic Editor